# Identifying Informational Sources in News Articles

Alexander Spangher[a,b], Nanyun Peng[c], Jonathan May[a,b] and Emilio Ferrara[a,b]

[a]Thomas Lord Department of Computer Science, University of Southern California
[b]Information Sciences Institute, University of Southern California
[c]University of California, Los Angeles

## Abstract

News articles are driven by the informational sources journalists use in reporting. Modeling when, how and why sources get used together in stories can help us better understand the information we consume and even help journalists with the task of producing it. In this work, we take steps toward this goal by constructing the largest and widest-ranging annotated dataset, to date, of informational sources used in news writing. We first show that our dataset can be used to train high-performing models for *information detection and source attribution*. Then, we introduce a novel task, source prediction, to study the *compositionality* of sources in news articles – i.e. how they are chosen to complement each other. We show good modeling performance on this task, indicating that there *is* a pattern to the way different sources are used *together* in news storytelling. This insight opens the door for a focus on sources in narrative science (i.e. planning-based language generation) and computational journalism (i.e. a source-recommendation system to aid journalists writing stories).[1]

## 1 Introduction

Journalism informs our worldviews; news itself is informed by the sources reporters use. Identifying sources of information in a news article is relevant to many tasks in NLP: misinformation detection (Hardalov et al., 2022), argumentation (Eger et al., 2017) and news discourse (Choubey et al., 2020).

Attributing information to sources is challenging: as shown in Table 1, while some attributions are identified via lexical cues (e.g. "said"), others are deeply implicit (e.g. one would have to know that ordering a "curfew" creates a public record that can be retrieved/verified). Previous modeling work, we show, has focused on the "easy" cases:

---

**News Article, *A***

Prime Minister **Laurent Lamothe** announced his resignation. ← *from* Statement

The announcement followed a corruption **commission**'s report. ← *from* Report

"There was no partisan intereference" said the **commission**. ← *from* Quote

However, curfews were imposed in cities in anticipation of protests. ← *from* Order

It remains to be seen whether the opposition will coalesce around a new candidate.

---

Table 1: Different informational sources used to compose a single news article. Source attributions shown in **bold**. Some sources may be implicit (e.g. 4th sent.) or too ambiguous (last sent.). Information types used by journalists are shown on the right. Our central question: *does this article need another source?*

identifying attributions via quotes,[2] resulting in high-precision, low recall techniques (Padó et al., 2019; Vaucher et al., 2021).

*In the first part of this paper* we address *source attribution*. We define the concept of "source" broadly to capture different information-gathering techniques used by journalists, introducing 16 categories of sourcing (some shown in Tables 1, 2, and 3). We apply this schema to construct the largest *source-attribution* dataset, to our knowledge, with 28,000 source-attributions in 1,304 news articles. Then, we train high-performing models, achieving an overall attribution accuracy of 83% by fine-tuning GPT3. We test numerous baselines and show that previous lexical approaches (Muzny et al., 2017), bootstrapping (Pavllo et al., 2018), and distant supervision

---

[1]All data and model code can be found at https://github.com/alex2awesome/source-exploration.

[2]By *quote*, we mean information derived from a person or a document – verbatim or paraphrased. *Sourced information* is broader and includes actions by the journalist to uncover information: first-person observations, analyses or experiments.

(Vaucher et al., 2021) fail.

*In the second part of this paper*, with source-attribution models in hand, we turn to a fundamental question in news writing: when and why are sources used together in an article? Sources tend to be used in canonical ways: an article covering local crime, for instance, will likely include quotes from both a victim and a police officer (Van Krieken, 2022; Spangher and Choudhary, 2022), and an article covering a political debate will include voices from multiple political parties (Hu et al., 2022). However, until now, the tools have not existed to study the *compositionality* of sources in news, i.e., why a set of sources was selected during the article's generative process.

To establish *compositionality*, we must show that a certain set of sources is *needed* in an article. We introduce a new task, *source prediction*: does this document need another source? We implement this task in two settings: (1) ablating news documents where all sentences attributable to a source are removed or (2) leveraging a dataset with news edit history (Spangher et al., 2022) where updates add sources. We show that large language models achieve up to 81.5% accuracy in some settings, indicating a degree of predictability. In doing so, we pave the way for downstream applications based on assumptions of *compositionality* in news, like source-based generative planning (Yao et al., 2019) and source recommendation engines (Spangher et al., 2021; Caswell, 2019).

In sum, our contributions are three-fold:

1. We introduce the first comprehensive corpus of news article source attribution, covering 1,304 news articles across an expansive set of 16 information channels.

2. We build state-of-the-art source attribution models, showing that they can be used to achieve 83% accuracy. Additionally, we validate that this task is challenging and requires in-depth comprehension by showing that fine-tuning GPT3 outperforms zero- and few-shot prompting, echoing results from (Ziems et al., 2023).

3. We open the door to further work in document-level planning and source-recommendation by showing news articles use sources in a *compositional* way. We show that models can predict when an article is missing a source 81.5% of the time.

A roadmap to the rest of the paper is as follows. In the next part, Section 2, we address our approach to *source attribution*: we define more precisely the ways in which a sentence is attributable to a source, the way we built our dataset, and our results. In Section 3, we discuss how we study *compositionality* in news writing by constructing a prediction problem. We close by discussing the implications and outlining future work.

## 2 Source Attribution

### 2.1 Problem Definition

We model a news article as a set of sentences, $S = \{s_1, ...s_n\}$ and a set of informational sources $Q = \{q_1, ...q_k\}$. We define an attribution function $a$ that maps each sentence to a subset of sources:[3]

$$a(s) \subseteq Q \text{ for } s \in S$$

A sentence *is* attributable to a source if there is an *explicit* or *implicit* indication that the facts in it came from that source. A sentence *is not* attributable to any source if the sentence does not convey concrete facts (i.e. it conveys journalist-provided analysis, speculation, or context), or if it cannot be determined where the facts originated.

Sources are people or organizations and are usually explicitly mentioned. They may be named entities (e.g. "Laurent Lamothe," in Table 1), or canonical indicators (e..g "commission," "authorities") and they are *not* pronouns. In some cases, a sentence's source is not mentioned in the article but can still be determined if (1) the information can only have come from a small number of commonly-used sources[4] or (2) the information is based on an eye-witness account by the journalist. See Table 2 for examples of these latter two categories. In the first two rows, we give examples of sourced information that a knowledgeable journalist could look up quickly. The third row shows a scene that could only have been either directly observed, either in-person or via recording, and thus must be sourced directly to the journalist.

We formulate this attribution task with two primary principles in mind: we wish to attribute as many sentences as possible to informational

---

[3]Most sentences are attributed to only one source in the article, but some are attributed to several.

[4]Examples in this category include "the stock market," "legislative/executive records," "court filings." Trained journalists can tell with relative accuracy where this information came from.

| Example sentences from different articles where sources are implicit |
|---|
| Tourist visits have declined, and the **Hong Kong stock market** has been falling for the past few weeks, but **protesters** called for more action.  ← `Published Work`, `Price Signal`, `Statement` |
| Mr. Trump was handed defeats in Pennsylvania, Arizona and Michigan, where a **state judge** in Detroit rejected an unusual Republican attempt to...  ← `Lawsuit` |
| Mr. Bannon, former chief strategist for President Trump, was warmly applauded when he addressed the party congress of the anti-immigrant National Front...  ← `Direct Observation` |

Table 2: Examples of sentences with sourced information that is non-obvious and not based on lexical cues. In the first two rows, we show sentences where sourcing is implicit but where a trained journalist can deduce the source. In the last row, we show a sourced sentence where the descriptive information could only have come from a direct observation by the journalist. **Bold names** are the source attribution, when it exists. In cases, not shown, where it does not exist, we label "passive voice". Underline indicates the specific information that was sourced. Colored annotations on the right are high-level information channels that could, in future work, be mined for source recommendations.

| Information Channel | Num. Sentences |
|---|---|
| No Quote | 23614 |
| Direct Quote | 7928 |
| Indirect Quote | 6564 |
| Background/Narrative | 3818 |
| Statement/Public Speech | 3280 |
| Published Work/Press Report | 2730 |
| Email/Social Media Post | 1352 |
| Proposal/Order/Law | 896 |
| Court Proceeding | 540 |
| Direct Observation | 302 |
| Other | 610 |

Table 3: Number of sentences in our corpus, according to the information channel by which the journalist observed the information.

sources used, and we wish to identify when the same source informed multiple sentences.[5] We allow for an expansive set of information channels to be considered (see Table 3 for some of the top channels) and design a set of 16 canonical information channels that journalists rely on.[6]

## 2.2 Corpus Creation and Annotation

We select 1,304 articles from the *NewsEdits* corpus (Spangher et al., 2022) and deduplicate across versions. In order to annotate sentences with their attributions, we recruit two annotators. One an-

notator is a trained journalist with over 4 years of experience working in a major newsroom, and the other is an undergraduate assistant. The senior annotator checks and mentors the junior annotator until they have a high agreement rate. Then, they collectively annotate 1,304 articles including 50 articles jointly. From these 50, we calculate an agreement rate of more than $\kappa = .85$ for source detection, $\kappa = .82$ for attribution. Categories shown in Table 3 are developed early in the annotation process and expanded until a reasonable set captures all further observations.[7] Categories are refined and adjusted following conversations with experienced journalists and journalism professors. For a full list of categories, see appendix[8].

## 2.3 Source Attribution Modeling

We split Source Attribution into two steps: *detection* (*is* the sentence attributable?) and *retrieval* (what is that attribution?) because, in early trials, we find that using different models for each step is more effective than modeling both jointly.

Prior work in Source Attribution primarily used hand-crafted rules (Peperkamp and Berendt, 2018), bootstrapping (Pavllo et al., 2018) and distance-supervision (Vaucher et al., 2021) approaches (see Section 4). Although such work has shown impressive performance on curated datasets, they typically define a source's informational contribution rather narrowly (i.e. only direct or indirect quotes, compared with the 16 channels listed in Table 3). So, we test several variations of

---

[5]For example, in Table 1, two sentences are attributable to the *commission*, despite the information coming from two separate channels (a published document and statement).

[6]These 16 categories are formulated both in conversation with journalists and after extensive annotation and schema expansion.

[7]We find $\kappa = .45$ agreement for quote-type categories

[8] ***Note:*** *we do not perform modeling on these categories in the present work, but use them for illustration and evaluation.*

| | | | Direct Quote | Indirect Quote | Statement/ Speech | Email/ Social | Published Work | Other | Micro Avg. |
|---|---|---|---|---|---|---|---|---|---|
| *Detection* f1 score | | Rules 1 | 64.7 | 69.3 | 81.2 | 76.2 | 72.7 | 37.4 | 59.1 |
| | | Rules 2 | 71.3 | 79.8 | 89.8 | 82.1 | 79.2 | 32.5 | 68.8 |
| | | Quootstrap | 85.0 | 81.3 | 51.3 | 58.6 | 33.1 | 3.0 | 33.4 |
| | | Sentence | 91.0 | 98.7 | 94.1 | **92.7** | 85.4 | 61.4 | 87.1 |
| | | Full-Doc | **92.0** | **98.7** | **96.4** | 89.8 | **86.4** | **65.1** | **88.2** |
| *Retrieval* Accuracy on gold-labeled sourced sents | | Rules 1 | 47.8 | 48.4 | 43.0 | 51.7 | 37.8 | 30.2 | 46.4 |
| | | +*coref* | 57.3 | 54.5 | 49.8 | 49.4 | 38.3 | 34.9 | 52.8 |
| | | Rules 2 | 20.7 | 22.5 | 30.3 | 21.3 | 27.4 | 30.2 | 22.5 |
| | | +*coref* | 31.6 | 42.0 | 56.1 | 30.3 | 32.3 | 30.2 | 36.6 |
| | | QuoteBank | 9.9 | 16.0 | 16.4 | 17.7 | 4.3 | 0.5 | 5.5 |
| | | SeqLabel | 37.2 | 43.4 | 40.0 | 31.2 | 32.3 | 17.7 | 38.5 |
| | | SpanDetect | 61.1 | 59.5 | 67.6 | 44.4 | 51.6 | 36.5 | 59.5 |
| | | +*coref* | 51.2 | 56.8 | 60.6 | 79.0 | 54.6 | 42.6 | 53.6 |
| | | GPT3 ft, Babbage | 80.9 | 86.9 | 85.0 | 71.9 | 57.9 | 38.3 | 78.9 |
| | | +*coref* | 78.7 | 82.5 | 76.3 | 56.1 | 54.4 | 31.2 | 73.2 |
| | | GPT3 ft, Curie | **94.0** | **95.5** | **91.1** | **91.0** | **81.6** | 57.3 | **91.4** |
| | | GPT3 0-shot, DaVinci | 70.9 | 58.8 | 72.5 | 43.1 | 54.6 | 47.6 | 58.5 |
| | | +*coref* | 66.9 | 57.6 | 61.9 | 20.2 | 42.6 | 51.4 | 55.4 |
| | | GPT3 few-shot, DaVinci | 74.9 | 56.5 | 70.1 | 52.3 | 49.4 | **82.8** | 61.6 |
| | | +*coref* | 70.0 | 55.6 | 72.7 | 50.5 | 48.8 | 60.7 | 58.6 |
| *Both* Acc all sents | | GPT3 ft, Babbage | 79.5 | 82.9 | 82.9 | 73.4 | 60.5 | 53.0 | 70.9 |
| | | +*Nones* | 82.4 | 84.8 | 85.9 | 73.4 | 61.0 | 64.5 | 73.1 |
| | | GPT3 ft, Curie | 90.4 | 90.7 | 89.9 | 91.1 | 78.0 | **68.9** | 80.0 |
| | | +*Nones* | **92.3** | **92.9** | **92.9** | **91.0** | **78.2** | 68.3 | **83.0** |

Table 4: *Detection*, or correctly identifying source sentences, *Retrieval* or correctly attributing sentences to sources, are two steps in *Source Attribution*. *Both* refers to the end-to-end process: first identifying that a sentence is a informed by a source *and then* identifying that source. +*coref* refers to performing coreference resolution beforehand, and universally hurts the model. +*None* refers to Retrieval models trained to assign "None" to sentences without sources, possibly eliminating false positives introduced by Detection. *Takeaway*: *We can attribute sources with accuracy* $> 80$.

methods introduced in prior work on our dataset to confirm that these categories are not implicitly attributed. For *detection*, a binary classification task, F1-score is used. For *retrieval*, we use accuracy, or precision@1.

**Baseline Methods**

*Rules 1 (R1): Co-Occurrence*: We identify sentences where a source entity candidate co-occurs with a speaking verb. For *detection*, any sentence that contains such a co-occurence is considered a detected sentence. For *attribution*, we consider the identity of the source entity. We use a list of 538 speaking verbs from Peperkamp and Berendt (2018) along with ones identified during annotation. We extract PERSON Named Entities and noun-phrase signifiers using a lexicon (n=300) (e.g. "authorities", "white house official") extracted from Newell et al. (2018)'s dataset.

*Rules 2 (R2): Governance*: Expanding on R1, we parse syntactic dependencies in sentences (Nivre, 2010) to introduce additional heuristics. Specifically, we identify sentences where the name

is an $nsubj$ dependency to a speaking verb governor. $nsubj$ is a grammatical part-of-speech, and a governor is a higher node in a syntactic parse tree.

*Quootstrap*: Pavllo et al. (2018) created a bootstrapping algorithm to discover lexical patterns indicative of sourcing. Contrasting with previous baselines, which hand-crafted lexical rules, bootstrapping allowed researchers to learn large numbers of highly specific patterns. Although the small size of our dataset compared with theirs prevents us from extracting novel lexical patterns tailored to us, we use a set of 1,000 lexical patterns provided by the authors[9]. Similary to R1 and R2, for *detection*, we consider all sentences that match these 1,000 lexical rules to be "detected" sentences. For *attribution*, we examine the entities these rules extract.

*QuoteBank*: In Vaucher et al. (2021), authors train a BERT-based entity-extraction model on distantly-supervised data scored from (Pavllo

[9] https://github.com/epfl-dlab/Quoteban k/blob/main/quootstrap/resources/seedPat terns.txt

et al., 2018). This method is less lexically focused, and thus more generalizable, than previous methods. They use their model to score and release a large corpus of documents. We examine this corpus and select articles that are both in their corpus and in our annotation set, finding 139 articles, and limit our evaluation to these articles. For *detection*, we examine all sentences with an attribution, and for *attribution*, we match the name of that source with our gold-labels.

### Detection Methods

*Sentence:* We adapt a binary sentence classifier where each token in each sentence is embedded using the `BigBird-base` transformer architecture (Zaheer et al., 2020). Tokens are combined via self attention to yield a sentence embedding and again to yield a document embedding. Thus, each sentence is independent of the others.

*Full-Doc:* We use a similar architecture to the Sentence approach, but instead of embedding tokens in each sentence separately, we embed tokens in the whole document, then split into sentences and combine using self-attention. Thus, the sentences are not embedded independently and are allowed to share information.

### Retrieval Methods

*Sequence Labeling*: predicts whether each token in a document is a source-token or not. We pass each document through `BigBird-base` to obtain token embeddings and then use a token-level classifier. We experiment with inducing a curriculum by training on shorter-documents first, and freezing layers 0-4 of the architecture.

*Span Detection*: predicts start and stop tokens of the sentence's source. We use `BigBird-base`, and separate start/stop-token classifiers (Devlin et al., 2018). We experiment with inducing decaying reward around start/stop positions to reward near-misses, and expand the objective to induce source salience as in Kirstain et al. (2021), but find no improvement.

*Generation*: We formulate retrieval as open-ended generation and fine-tune GPT3 models to generate source-names. We use with the following prompt: "`<article>To which source can we attribute the sentence <sentence>?`". We need to include the whole article in order to capture cases where a source is mentioned in another sentence. We experiment with fine-tuning Babbage and Curie models, and testing zero- and few-shot for DaVinci models.

|  | Gold (Train) | Gold (Test) | Silver |
|---|---|---|---|
| # docs | 1032 | 272 | 9051 |
| # sent / doc | 30 | 67.5 | 27 |
| doc len (chars) | 3952 | 7885 | 3984 |
| # sources / doc | 6.8 | 12.1 | 8.2 |
| % sents sourced | 47.7% | 46.9% | 57.4% |
| % sents, most-used source / doc | 37.5% | 28.1% | 31.8% |
| % sents, least-used source / doc | 5.9% | 2.4% | 6.7% |
| source entropy | 1.6 | 2.1 | 1.8 |
| # sources added per version | n/a | n/a | +2 |
| document sent. ↑ likely to be sourced | 96th p | 92th p | 0th p |

Table 5: Corpus-level statistics for our training, test, and silver-standard datasets. Shown are averages across the entire corpus. Documents in the test set are longer than the training, but the model seems to generalize well to the silver-standard corpus, as statistics match. "% sents, top source" and "% sents, bot source" refer to the % of sourced sentences attributed to the most- and least-used sources in a story. "# sources added / version" shows the number of sources added to articles each news update; it is calculated using the NewsEdits corpus (Spangher et al., 2022). "sentence most likely to be sourced" refers to the sentence with the highest likelihood of being a sourced sentence, as a percentile of doc. length

Because our prompt-query as it contains an entire article/source pair, we have limited additional token-budget; so, for our few-shot setting, we give examples of sentence/source pairs where the source is mentioned in the sentence.

For *+coref* variations, we evaluate approaches on articles after resolving all coreferences using LingMess (Otmazgin et al., 2022). For *+Nones* variations, we additionally train our models to detect when sentences do *not* contain sources. We use this as a further corrective to eliminate false positives introduced during detection.

### 2.4 Source Attribution Results

As shown in Table 4, we find that the GPT3 Curie source-retrieval model paired with the Full-Doc detection module in a pipeline performed best, achieving an attribution accuracy of 83%. In the *+None* setting, both GPT3 Babbage and Curie can identify false positives introduced by the detection stage and outperform their counterparts. Overall, we find that resolving coreference does not improve performance, despite similarities between

the tasks.

The poor performance of both rules-based approaches and QuoteBank, which also uses heuristics,[10] indicates that simple lexical cues are insufficient. Although QuoteBank authors reported it outperformed similar baselines as we tested (Vaucher et al., 2021), we observe low performance from Quotebank (Vaucher et al., 2021), even in categories it is trained to detect.

GPT3 DaVinci zero-shot and few-shot greatly underperform fine-tuned models in almost all categories (except "Other"). Further, we see very little improvement in the use of a few-shot setup vs. zero-shot. This might be because the examples we give GPT3 are sentence/source pairs, which do not correctly mimic our document-level source-attribution task. We face shortcomings due to the document-level nature of our task: the token-budget required to ask a document-level question severely limits our ability to do effective few-shot document-level prompting. Approaches that condense prompts (Mu et al., 2023) might be helpful to explore in future work.

## 2.5 Insights from Source Analysis

Having built an attribution pipeline that performs reasonably well, we run our best-performing attribution model across 9051 unlabeled documents from *NewsEdits* and extract all sources. In this section, we explain derive insights into how sources are used in news articles. For statistics guiding these insights, see in Table 5, which shows statistics calculated on both our annotated dataset ("Gold Train" and "Gold Test" columns) and the 9051 documents we just described ("Silver" column). We ask two primary questions: *how much an article is sourced?* and *when are sources used in the reporting and writing process?*

**Insight #1:** $\sim 50\%$ **of sentences are sourced, and sources are used unevenly.** Most articles, we find, attribute roughly half the information in their sentences to sources. This indicates that the percentage of sources used is fairly consistent between longer and shorter documents. So, as a document grows, it adds roughly an equal amount of sourced and unsourced content (e.g. explanations, analysis, predictions).[11] We also find that sources

are used unevenly. The most-used source in each article contributes $\sim 35\%$ of sourced sentences, whereas the least-used source contributes $\sim 5\%$. This shows a hierarchy between major and minor sources used in reporting and suggests future work analyzing the differences between these sources.

**Insight #2: Sources begin and end documents, and are added while reporting** Next we examine when sources are used in the reporting process. We find that articles early in their publication cycle tend to have fewer sources, and add on average two sources per subsequent version. This indicates an avenue of future work: understanding which kinds of sources get added in later versions can help us recommend sources as the journalist is writing. Finally, we also find, in terms of narrative structure, that journalists tend to lead their stories with sourced information: the most likely position for a source is the first sentence, the least likely position is the second. The second-most likely position is the end of the document.[12] *A caveat to Table 5*: many gold-labeled documents were parsed so the first sentence got split over several sentences, which is why we observe the last sentences having highest sourcing.[13]

## 3 Source Compositionality

### 3.1 Problem Definition

Our formulation in Section 2 for quotation attribution aims to identify the set of sources a journalist used in their reporting. Can we reason about why certain groups of sources were chosen in tandem? Can we determine if an article is *missing* sources?

We create two probes for this question:

1. *Ablation*: Given an article $(S, Q)$ with attribution $a(s) \forall s \in S$, choose one source $q \in Q$. To generate positive examples, remove all sentences $s$ where $q \in a(s)$. To generate negative, remove an equal number of sentences where $a(s) = \{\}$ (i.e. no source).

2. *NewsEdits*: Sample article-versions from *NewsEdits*, a corpus of news articles with their updates across time (Spangher et al.,

---

[10]Quotebank's algorithm condenses input data to a BERT span-classifier by (1) looking for double-quotes (2) identifying candidate speakers through a lookup table.

[11]For more details, see the appendix.

[12]The sources might be used for different purposes: Spangher et al. (2023) performed an analysis on news articles' narrative structure, and found that sentences conveying the *Main Idea* lead the article while sentences conveying *Evaluations* or *Predictions*.

[13]E.g. `sents=['BAGHDAD', '--', 'Yesterday, the American military said']`. See appendix, Figure 4.

| | | Other News | Disaster | Elections | Labor | Safety |
|---|---|---|---|---|---|---|
| **Top Ablated** | FastText | 66.1 | 65.8 | 69.8 | 68.8 | 68.0 |
| | +*Source-Attribution* | 66.0 | 64.5 | 69.8 | 68.2 | 68.0 |
| | BigBird | 74.2 | 68.4 | 78.3 | **74.0** | 78.1 |
| | +*Source-Attribution* | 73.9 | 69.7 | 74.9 | 73.4 | 73.4 |
| | GPT3 ft, Babbage | **78.3** | **75.5** | **81.5** | 72.7 | **80.0** |
| | +*Source-Attribution* | 74.9 | 69.5 | 78.0 | 70.9 | 65.1 |
| **Second Source** | FastText | 57.6 | 63.2 | 60.8 | 61.0 | 63.3 |
| | +*Source-Attribution* | 57.8 | 63.2 | 61.1 | 62.3 | 64.1 |
| | BigBird | 63.8 | 61.8 | 63.1 | 64.3 | 61.7 |
| | +*Source-Attribution* | 65.1 | **69.7** | 65.7 | 64.9 | 62.5 |
| | GPT3 ft, Babbage | **67.1** | 67.9 | **72.9** | 58.8 | 65.6 |
| | +*Source-Attribution* | 65.4 | 65.1 | 68.0 | **65.9** | **66.7** |
| **Any Source** | FastText | 54.5 | 60.5 | 57.1 | 57.8 | 56.2 |
| | +*Source-Attribution* | 54.8 | **59.2** | 57.6 | 56.5 | 56.2 |
| | BigBird | 57.5 | 53.9 | 55.5 | 55.8 | **57.8** |
| | +*Source-Attribution* | **59.4** | 55.3 | 60.6 | 60.4 | 56.2 |
| | GPT3 ft, Babbage | 55.0 | 53.9 | **63.6** | 63.4 | 49.0 |
| | +*Source-Attribution* | 59.0 | 56.1 | 61.3 | 39.3 | 51.7 |
| **News Edits** | FastText | 58.1 | 48.9 | 62.1 | 58.6 | 48.8 |
| | +*Source-Attribution* | 56.8 | 55.8 | 61.9 | 61.2 | 49.6 |
| | BigBird | 63.5 | 63.9 | 64.5 | **64.8** | 64.8 |
| | +*Source-Attribution* | **69.4** | **65.3** | 62.6 | 60.4 | **64.2** |
| | GPT3 ft, Babbage | 65.0 | 63.9 | **64.6** | 62.4 | 51.0 |
| | +*Source-Attribution* | 64.0 | 56.1 | 61.3 | 39.3 | 51.7 |

Table 6: Results for *Source Prediction*, broken into four canonical news topics and 'other.' "Top Ablated" is our prediction task run on articles ablated by removing the source that has the most sentences, "Second Source" is where a source contributing more than 10% of sentences is removed, and "Any Source" is where any source is randomly removed. The *NewsEdits* task is to predict whether the article at time $t$ will be added sources at time $t + 1$. In the +*Source-Attribution* experiments, we add sourcing information, derived in Section 2, to the input (see Section 3.2). *Takeaway: On all of these tasks, our models were able to significantly outperform random (50% acc.). In general, our expectations are confirmed that: (a) harder tasks yield lower-accuracy results and (b) more powerful models improve performance. This indicates that there is a pattern how sources are used in news writing.*

2022). Identify articles at time $t$ where the update at $t + 1$ either adds a source or not.

Each probe tests source usage in different ways. *Ablation* assumes that the composition of sources in an article is cohesively balanced, and induces reasoning about this balance. *NewsEdits* relaxes this assumption and probes if this composition might change, either due to the article's completeness, changing world events that necessitate new sources, or some other factor.[14]

### 3.2 Dataset Construction and Modeling

We use our *Source Attribution* methods discussed in Section 2 to create large silver-standard datasets in the following manner for our two primary experimental variants: *Ablation* and *NewsEdits*. To interpret results in each variant better, we train a classifier to categorize articles into four topics plus one "other" topic[15], based on articles in the *New York Times Annotated Corpus* (Sandhaus, 2008) with keyword sets corresponding to each topic.

**Ablation** We take 9051 silver-standard documents (the same ones explored in Section 2.5) and design three variations of this task. As shown in Table 5, articles tend to use sources lopsidedly: one source is usually primary. Thus, we design Easy (Top Source, in Table 1), Medium (Secondary) and Hard (Any Source) variations of our task. For Easy, we choose the source with the most sentences attributed to it. For Medium, we randomly choose among the top 3 sources. And for Hard, we randomly choose any of the sources. Then, we create a $y = 1$ example by removing all sentences attributed to the chosen source, and we create a $y = 0$ example from the same document by removing an equal number of sentences that are

[14]Spangher et al. (2022) found that many news updates were factual and tied to event changes, indicating a breaking news cycle.

[15]These four have been identified as especially socially valuable topics, or "beats," due to their impact on government responsiveness (Hamilton, 2011)

not attributed to any sources.

**NewsEdits**   We sample an additional $40,000$ articles from the *NewsEdits* corpora and perform *attribution* on them. We sample versions pairs that have roughly the same number of added, deleted and edited sentences in between versions in order to reduce possible confounders, as Spangher et al. (2022) showed that these edit-operations were predictable. We identify article-version pairs where 2 or more sources were added between version $t$ and $t+1$ and label these as $y=1$, and 0 or 1 sources added as $y=0$.

**Modeling**   We use three models: (1) FastText (Joulin et al., 2016) for sentence classification, (2) A BigBird-based model: we use BigBird with self-attention for document classification, similar to Spangher et al. (2022).[16] Finally, (3) we fine-tune GPT3 Babbage to perform prompt-completion for binary classification. For each model, we test two setups. First, we train on the vanilla text of the document. Then, in the *+Source-Attribution* variants, we train by appending each sentence's source *attribution* to the end of it.[17] The source annotations are obtained from our attribution pipeline.

### 3.3   Results and Discussion

The results in Table 6 show that we are broadly able to predict when major sources (Top, Secondary) are removed from articles, indicating that there is indeed *compositionality*, or intention, in the way sources are chosen to appear together in news articles. The primary source (Top)'s absence is the easiest to detect, indicating that many stories revolve around a single source that adds crucial information. Secondary sources (Second) are still predictable, showing that they serve an important role. Minor sources (Any)'s absence are the hardest to predict and the least crucial to a story. Finally, source-addition across versions (NewsEdits) is the hardest to detect, indicating that versions contain balanced compositions.

---

[16]Concretely, we obtain token embeddings of the entire document, which we combine for each sentence using self-attention. We contextualize each sentence embedding using a shallow transformer architecture. We finally combine these sentence embeddings using another self-attention layer to obtain a document embedding for classification. We utilize curriculum learning based on document length, a linear loss-decay schedule.

[17]Like so: `<sent 1>. SOURCE: <source 1>. <sent 2> SOURCE: <source 2>... <sent n> SOURCE: <source n>`.

Overall, we find that our experiments are statistically significant from random (50% accuracy) with t-test $p < .01$, potentially allowing us to reject the null hypothesis that positive documents are indistinguishable from negative in both settings. Statistical significance does not preclude confounding, and both the *Ablation* and the *NewsEdits* setups contain possible confounders. In the *Ablation* set up, we might be inadvertently learning stylistic differences rather than source-based differences. To reduce this risk, we investigate several factors. First, we consider whether lexical confounders, such as speaking verbs, might be artificially removed in the ablated documents. We use lexicons defined in our rules-based methods to measure the number of speaking verbs in our dataset. We find a mean of $n = [34, 32]$ speaking verbs per document in $y = [0, 1]$ classes in the Top case, $n = [35, 34]$ in the Medium, and $n = [35, 37]$ in Hard. None of these differences are statistically significant. We also do not find statistically significant differences between counts of named entities or source signifiers (defined in Section 4). Finally, we create secondary test sets where $y = 0$ is non-ablated documents. This changes the nature of the stylistic differences between $y = 1$ and $y = 0$ while not affecting sourcing differences[18]. We rerun trials in the *Top* grouping, as this would show us the greatest confounding effect, and find that the accuracy of our classifiers differs by within -/+3 points.

In the *NewsEdits* setup, we have taken care to balance our dataset along axes where prior work have found predictability. For instance, Spangher et al. (2022) found that an edit-operations[19] could be predicted. So, we balance for length, version number and edit operations.

Having attempted to address confounding in various ways in both experiments, we take them together to indicate that, despite each probing different questions around sourcing, there are patterns to the way sources are during the journalistic reporting process. To illustrate, we find in Table 6 that Election coverage is the most easily predictable across all tasks. This might be because of efforts to include both left-wing and right-wing voices. It also might be because the cast of charac-

---

[18]We do not want to *train* on such datasets, because there are statistically significant length differences and other stylistic concerns ablated and non-ablated articles.

[19]E.g. Whether a sentence would be added in a subsequent version.

ters (e.g. campaign strategists, volunteers, voters) stays relatively consistent across stories.

Two additional findings are that (1) the tasks we expect are harder do yield lower accuracies and, (2) larger GPT3-based language models generally perform better. Although not especially surprising, it further confirms our intuitions about what these tasks are probing. We were surprised to find that, in general, adding additional information in both stages of this project, whether coreference in the *attribution* stage or source information in the *prediction* stage, did not improve the models' performance. (In contrast, adding source information to smaller language model, `BigBird`, helped with harder tasks like the Medium, Hard and *NewsEdits*). We had hypothesized that the signal introduced by this labeling would not harm the GPT3-based models, but this was not the case. It could be that the larger models are already incorporating a notion of coreference and attribution, and adding this information changed English grammar in a way that harmed performance.

## 4 Related Work

**Quote Attribution** Prior work in quote attribution has also been aimed at identifying which sources contributed information in news articles. Early work explored rules-based methods (Elson and McKeown, 2010; O'Keefe et al., 2012) and statistical classifiers (Pareti et al., 2013) to attribute sources to quotes. More recent work has extended these ideas by using bootstrapping to discover new patterns, *Quootstrap* (Pavllo et al., 2018) and training BERT-based models on perturbations on these patterns, or *QuoteBERT* (Vaucher et al., 2021). One upside of *Quootstrap* and *QuoteBERT* is that they might adapt better to new domains by learning and generalizing from new patterns. However, the method by which patterns are learned, finding quotes that repeat across outlets, might bias this method towards discovering quotes by oft-quoted figures. These quotes, in turn, may be contextualized differently than other quotes, introducing fundamental biases in which sources get discovered. We urge more consideration of these potential biases, not only for performance considerations but fairness. Overall, our work differs from previous work in this field because we defined *information* more broadly. Prior work is quote-focused, whereas we include a larger set of information channels (Table 3).

**Persona Modeling** A second area that our work draws inspiration from is the study of narrative characters and how they are used in fiction. Work by Bamman et al. (2013) and Card et al. (2016) used custom topic models to model characters by latent "personas" generated from latent document-level distributions.Earlier work extended this topic-modeling approach to news sources (Spangher et al., 2021). We see potential for future work merging this with our dataset and framework, using methods like discrete variational autoencoders, which have been applied to document-planning (Ji and Huang, 2021).

**Downstream Applications** : **Diversity** An interesting downstream applications of our work is to improve analysis of diversity in sourcing. Source-diversity has been studied in news articles (Peperkamp and Berendt, 2018; Masini et al., 2018; Berendt et al., 2021; Amsalem et al., 2020), where authors have constructed ontologies to further explore the role of sources from different backgrounds. **Opinion Mining** Another strain focuses on characterizing voices in a text by opinion (O'Keefe et al., 2013). Such work has been applied in computational platforms for journalists (Radford et al., 2015) and in fake news detection (Conforti et al., 2018).

## 5 Conclusions

We have offered a more expansive definition of sourcing in journalism and introduced the largest attribution dataset capturing this notion. We have developed strong models to identify and attribute information in news articles. We have used these attribution models to create a large silver standard dataset that we used to probe whether source inclusion in news writing follows predictable patterns. Overall, we intend this work to serve as a starting point for future inquiries into the nature of source inclusion in news articles. We hope to improve various downstream tasks in NLP and, ultimately, take steps towards building a source recommendation engine that can help journalists in the task of reporting.

## 6 Acknowledgments

Alexander Spangher would like to thank Bloomberg News for a generous 4 year fellowship that has funded this work and his other work in the areas of computational journalism, computational law, nuclear fusion, and linguistics.

# 7 Limitations

A central limitation to our work is that the datasets we used to train our models are all in English. As mentioned previously, we used English language sources from Spangher et al. (2022)'s *NewsEdits* dataset, which consists of sources such as nytimes.com, bbc.com, washingtonpost.com, etc.

Thus, we must view our work in source extraction and prediction with the important caveat that non-Western news outlets may not follow the same source-usage patterns and discourse structures in writing their news articles as outlets from other regions. We might face extraction biases if we were to attempt to do such work in other languages, such as only extracting sources that present in patterns similar to those observed in Western sources, which should be considered as a fairness issue.

# 8 Ethics Statement

## 8.1 Risks

Since we constructed our datasets on well-trusted news outlets, we assumed that every informational sentence was factual, to the best of the journalist's ability, and honestly constructed. We have no guarantees that such an attribution system would work in a setting where a journalist was acting adversarially.

There is a risk that, if such a work were used in a larger news domain, it could fall prey to attributing misinformation or disinformation. Thus, any downstream tasks that might seek to gather sourced sentences might be poisoned by such a dataset. This risk is acute in the news domain, where fake news outlets peddle false stories that attempt to *look* true (Boyd et al., 2018; Spangher et al., 2020). We have not experimented how our classifiers would function in such a domain. There is work using discourse-structure to identify misinformation (Abbas, 2022; Sitaula et al., 2020), and this could be useful in a source-attribution pipeline to mitigate such risks.

We used OpenAI Finetuning to train the GPT3 variants. We recognize that OpenAI is not transparent about its training process, and this might reduce the reproducibility of our process. We also recognize that OpenAI owns the models we finetuned, and thus we cannot release them publicly. Both of these thrusts are anti-science and anti-openness and we disagree with them on principle. However, their models are still useful in a black-box sense for giving strong baselines for predictive problems and drawing scientific conclusions about hypotheses.

## 8.2 Licensing

The dataset we used, *NewsEdits* (Spangher et al., 2022), is released academically. Authors claim that they received permission from the publishers to release their dataset, and it was published as a dataset resource in NAACL 2023. We have had lawyers at a major media company ascertain that this dataset was low risk for copyright infringement.

## 8.3 Computational Resources

The experiments in our paper required computational resources. We used 8 40GB NVIDIA V100 GPUs, Google Cloud Platform storage and CPU capabilities. We designed all our models to run on 1 GPU, so they did not need to utilize model or data-parallelism. However, we still need to recognize that not all researchers have access to this type of equipment.

We used Huggingface `Bigbird-base` models for our predictive tasks, and will release the code of all the custom architectures that we constructed. Our models do not exceed 300 million parameters.

## 8.4 Annotators

We recruited annotators from our educational institutions. They consented to the experiment in exchange for mentoring and acknowledgement in the final paper. One is an undergraduate student, and the other is a former journalist. Both annotators are male. Both identify as cis-gender. The annotation conducted for this work was deemed exempt from review by our Institutional Review Board.

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

## A Exploratory Data Analysis

We show more data analysis around source usage in news articles. Figure 1 shows the distribution over the amount of sentences in each article that are attributable to sources. Although most articles have around 50% of their sentences as source sentences, a small number of articles source $< 10\%$ of their sentences (a manual analysis shows that that these are mainly opinion pieces) or $> 90\%$ of their sentences (a manual analysis shows that these are mainly short, one-paragraph breaking news excerpts).

Figure 2 shows how articles grow over time, through versions. We find that, on average, two sources are added per version. This is surprisingly linear, with earlier versions containing the least number of sources.

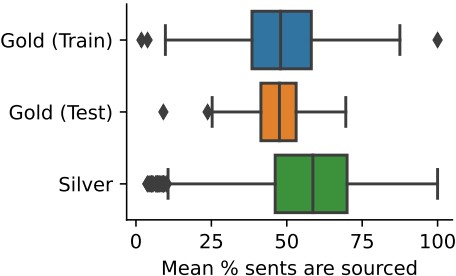

Figure 1: How much of a document is sourced? We show % of sourced sentences in documents.

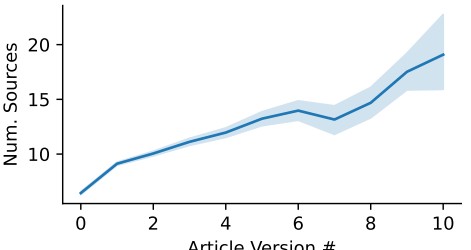

Figure 2: Do more sources get added to an article over time? We show the number of sources in an article as it gets republished, based on *NewsEdits* (Spangher et al., 2022) and find that as news unfolds, sources get added.

In Figure 3, we show that the percent of sourcing is consistent the longer a document gets. This means that when more sentences are added to the document, the journalists adds a consistent amount of sourced and non-sourced sentences. The only exception is when articles are very short. Manual inspection reveals that these are usually breaking news paragraphs that are entirely composed of a reference to a press release, a statement or a quote.

In Figure 4, we show the likelihood of a source being present in each sentence-position of our document. This indicates where in the document sources are used. The likeliest spot for a source is the first sentence, and the least likely is the second sentence. As can be seen, the likelihood of a source increases further throughout the document.

## B  Annotation Definitions

1. **Quote**: A statement or a paraphrase of a statement given by the source to the reporter in an interview.

2. **Background:** A sentence giving non-event information the source (i.e. descriptions, role, status, etc.), that does *not* contain a

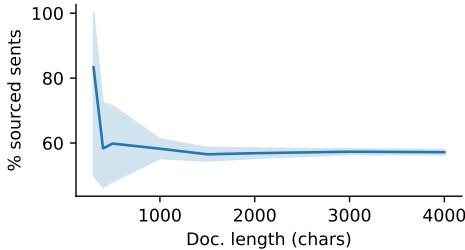

Figure 3: Do longer stories contain more sourced information? We show the percentage of sentences in an article that are based on sourced information based on the length of the document. Shorter stories are almost entirely composed of sourced sentences.

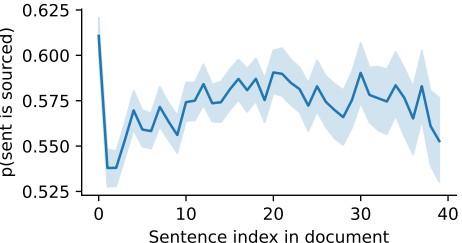

Figure 4: Where in the article are sources used? We show the likelihood of a sentence is sourced based on its position within an article. Sources are most likely to be used in the first sentence and least likely to be used in the second.

quote. Does not have to contain any external source of information.

3. **Narrative:** A sentence giving narrative information about the source's role in events that does *not* contain a quote. Does not have to contain any external source of information.

   (a) For "Background" and "Narrative," these usually don't explicitly reference external sources of information. It's typically implied that the journalist learned this information by talking to the sources, but it does not have to be the case.

4. **Direct Observation:** A sentence where it's clear that the journalist is either (1) literally witnessing the events (2) conducting their own analysis, investigation or experiment, i.e. the journalist is their own source of *observation*.

   (a) The difference between "Narrative" and "Direct Observation" can be hazy. Unless it is very clear that the journalist

is literally observing events unfold, do NOT use "Direct Observation."

  (b) When "Direct Observation" is selected, the source head is "journalist," source type is "Named Individual," affiliation is "Media," role is "informational" and status is "current," UNLESS the journalist abundantly defines themselves as otherwise (i.e. "In my years as a diplomat...," "I am a professor...").

5. **Public Speech**: Remarks made by the source in a public setting to an open crowd.

6. **Communication**: Remarks made by the source in a private setting or to a closed, select group. Can be interpreted broadly to include written communications.

7. **Published Work**: Work written by the source, usually distributed via academic journals or government publications.

8. **Statement:** A prepared quote given by a source. Usually distributed in a press conference or in writing.

9. **Lawsuit:** Any information given during the course of a court proceeding including claims, defense, rulings or other court-related procedures.

10. **Price Signal:** Any information about a company's stock price, the price of goods, etc. that was obtained through analyzing market data.

11. **Vote/Poll**: Information given about voting decisions, whether as a result of an actual vote or a electoral or opinion poll.

12. **Document**: A more generic category of information distributed via writing (**Published Work** is a subset of this class).

13. **Press Report**: Information obtained from a media source, whether it's a news article, a television report or a radio report.

14. **Social Media Post**: Information posted on a social media platform (i.e. Twitter, Facebook, blog comments, etc.).

15. **Proposal/Order/Law**: Information codified in text by officials resulting, or intending to result in, policy changes (i.e. executive order, legislative text, etc.).

16. **Declined Comment**: A special category of quote where the source does not comment. Also includes when a source "could not be reached for comment."