# OpenReview forum: "Identifying Informational Sources in News Articles"
_EMNLP/2023/Conference — EMNLP 2023 Main_

### Official Review · Reviewer_6Nn9 · 2023-07-24

**Soundness:** 4

**Excitement:**

4: Strong: This paper deepens the understanding of some phenomenon or lowers the barriers to an existing research direction.

**Paper Topic And Main Contributions:**

The paper focuses on the task of source attribution, information detection, and the novel task of source prediction. To study the compositionality of sources in news articles, a dataset is constructed and shared with news articles annotated with their source and corresponding information type. After training/evaluating several models on source detection/retrieval and thorough analysis and insights, the focus shifts to Source Compositionality. Meaning, “Can we determine if an article is missing sources?”. After a series of ablation studies, the result is, that it can be done though difficult.

**Reasons To Accept:**

A new task is introduced together with an annotated and silver dataset.
Several interesting insights when it comes to news and how they are structured/sourced.
Ablation studies, in-depth analysis of results, and insights provide added value to the community.

**Reasons To Reject:**

None.

**Reproducibility:**

3: Could reproduce the results with some difficulty. The settings of parameters are underspecified or subjectively determined; the training/evaluation data are not widely available.

**Reviewer Confidence:**

3: Pretty sure, but there's a chance I missed something. Although I have a good feel for this area in general, I did not carefully check the paper's details, e.g., the math, experimental design, or novelty.

**Typos Grammar Style And Presentation Improvements:**

Table 6: In “News Edits/BigBird/Safety” the wrong value is highlighted
539: “W” at the end of the paragraph.
894-897: “information the source” is wrong. “information of the source”?
901: Latex Error around “not”

---

> ### Author Rebuttal · Authors · 2023-08-29
>
> Thank you so much for your thoughtful review and kind comments!!

---

### Official Review · Reviewer_d5Gh · 2023-08-12

**Soundness:** 3

**Excitement:**

4: Strong: This paper deepens the understanding of some phenomenon or lowers the barriers to an existing research direction.

**Paper Topic And Main Contributions:**

The paper focuses on the nature of news articles. More specifically, it introduces 16 different sources that are used by journalists in writing articles. The authors employ two human annotators to identify the sources used for each sentence of 1304 different news articles. Moreover, the annotators mark the tokens that refer to the source to which each sentence can be attributed. The paper then compares baseline and more advanced models to determine whether a sentence is sourced from an informational channel or not, in addition to finding the source of the information. These models are then applied to a larger set of articles, demonstrating how the models can be used to uncover information about the number of sources per article, and the composition of sources in an article.

**Questions For The Authors:**

1) The First contribution (Lines 86-89) mentions 13 informational categories, however, 16 different categories are listed in the abstract. Is that a typo?
2) Do you think there is a substantial difference between the articles from different news topics shown in Table 6? If so, then I would recommend moving the paragraph describing the categorization using distant supervision (starting at Line 399) to the beginning of Subsection §3.2, as it currently is under modelling, so I expected it to be a new model.

**Reasons To Accept:**

* The paper can introduce new applications of NLP in the field of journalism.
* Source attribution allows for a deeper analysis of the characteristics of news articles.
* The authors attempted to provide possible explanations for the results, especially for the second experiment in §3, which I find to be useful.

**Reasons To Reject:**

* The paper could benefit from another round of editing. While I do not think it is a major weakness, I believe it will make the paper easier to follow, and understand for the reader.
* I think the description of some models is unclear, which might make them harder to replicate. For example, the baseline R1 (Lines 182-191) mentions the terms source entity candidate and speaking verbs. It was not until the second reading of the paragraph that I thought I had an adequate understanding of what they mean.

**Reproducibility:**

3: Could reproduce the results with some difficulty. The settings of parameters are underspecified or subjectively determined; the training/evaluation data are not widely available.

**Reviewer Confidence:**

3: Pretty sure, but there's a chance I missed something. Although I have a good feel for this area in general, I did not carefully check the paper's details, e.g., the math, experimental design, or novelty.

**Typos Grammar Style And Presentation Improvements:**

* The Limitations section should be a separate section and not a subsection for the “Ethics Statement” section.
* I got a bit confused on what you mean by “informational source” (Title), “informational category” (Line 89) and “information channel” (Table 3). While I guess the last two refer to the same concept, it would be easier for the reader if you use the same term for it.
* Typos and grammatical errors:
  * Line 230: We prompts
  * Line 506-507: Such works focuses
  * Line 518-519: Both our work probes how characters in text relate.
  * Line 539: W

---

### Official Review · Reviewer_5ZFc · 2023-08-12

**Soundness:** 3

**Excitement:**

4: Strong: This paper deepens the understanding of some phenomenon or lowers the barriers to an existing research direction.

**Paper Topic And Main Contributions:**

The paper is about trying to identify the set of sources journalists use when reporting by way of a function that maps news article candidate sentences to corresponding sources. This task is done in two-steps: first detecting which sentences are “attributable” (i.e linkable to sources), then retrieving the attributed source. After identifying which sources each news article sentence is attributed to, authors further developed a task which aims at predicting which sources news articles might be missing.

**Reasons To Accept:**

- First of its kind corpus of news article source attributions, covering 1,304 news articles across various informational categories (e.g press report, social media post)
- The motivation for this research area is strong (e.g developing tools to aid journalists review/write articles in a source-based manner). The source attribution corpus is a first step in that direction.

**Reasons To Reject:**

 - No direct comparison to existing benchmarks of existing similar tasks (e.g literature on tasks such as cite-worthiness detection, quote attribution, source prediction). It is not obvious, for any of the two tasks the paper formulates, to understand how the new methods/models contributed by this paper rank in comparison to existing work in the research area.

**Reproducibility:**

3: Could reproduce the results with some difficulty. The settings of parameters are underspecified or subjectively determined; the training/evaluation data are not widely available.

**Reviewer Confidence:**

3: Pretty sure, but there's a chance I missed something. Although I have a good feel for this area in general, I did not carefully check the paper's details, e.g., the math, experimental design, or novelty.

---

> ### Author Rebuttal · Authors · 2023-08-29
>
> Thank you so much for your thoughtful review! We ask you to please reconsider this one point:
>
> > No direct comparison to existing benchmarks of existing similar tasks (e.g literature on tasks such as cite-worthiness detection, quote attribution, source prediction).
>
> We do try to compare against extensive existing methods for quote attribution. R1 and R2 are derived from [1]'s work. Quootstrap and Quotebank are directly derived from [2]s and [3]s work. We include description in 181-195 explicitly referencing these.
>
> [1] Jeroen Peperkamp and Bettina Berendt. 2018. Diver- sity checker: Toward recommendations for improv- ing journalism with respect to diversity. In Adjunct Publication of the 26th Conference on User Model- ing, Adaptation and Personalization, pages 35–41.
> [2] Dario Pavllo, Tiziano Piccardi, and Robert West. 2018. Quootstrap: Scalable unsupervised extraction of quotation-speaker pairs from large news corpora via bootstrapping. In Twelfth International AAAI Con- ference on Web and Social Media.
> [3] Timote Vaucher, Andreas Spitz, Michele Catasta, and Robert West. 2021. Quotebank: a corpus of quotations from a decade of news. In Proceedings of the 14th ACM International Conference on Web Search and Data Mining, pages 328–336.
>
> It's true that we did not compare to the training and test sets that these authors used in their works, but there isn't really a single established benchmark for Quote Attribution as a task. Even when follow-up work comes from the same lab, as in [2], [3]. authors typically stated and redid the benchmark.
>
> We believe that the benchmarks created by prior authors are limited, for reasons we discuss in the Introduction and Related Works. But to be explicit, we focused on Informational Sources that are introduced via several lexical techniques, not just direct quotes, as is typically introduced in prior work ( we discuss in line 509-511 how prior work even fails for Direct Quotations).
>
> Thus, our techniques really shine in new areas of source-usage that aren't typically covered by prior benchmarks. We can do a better job explaining this in the camera ready.

---

### Meta-Review · Area_Chair_gN6N · 2023-09-23

**Recommendation:** 5

**Metareview:**

The paper studies how to identify and analyze the sources that journalists use in their news articles. The authors defined 16 types of sources and annotated 1304 news articles with the source type and the parts of the article referring to the source. The paper then develops and evaluates models to detect whether a sentence can be attributed to a source or not, and to retrieve the source. The paper also applies the model to a larger corpus of news articles to reveal insights about different reference patterns.

Overall, the paper introduces a new task and contributes an important dataset for further research. All three reviewers appreciated the novelty and the potential impact of the work. However, they also provide some suggestions to further improve the readability of the paper which I would encourage the authors to consider while preparing the future version of the paper.

---

### Decision · Program_Chairs · 2023-10-07

**Decision:**

Accept-Main

**Comment:**

The paper studies how to identify and analyze the sources that journalists use in their news articles. The authors defined 16 types of sources and annotated 1304 news articles with the source type and the parts of the article referring to the source. The paper then develops and evaluates models to detect whether a sentence can be attributed to a source or not, and to retrieve the source. The paper also applies the model to a larger corpus of news articles to reveal insights about different reference patterns.

Overall, the paper introduces a new task and contributes an important dataset for further research. All three reviewers appreciated the novelty and the potential impact of the work. However, they also provide some suggestions to further improve the readability of the paper which I would encourage the authors to consider while preparing the future version of the paper.